# Giant capsids from lattice self-assembly of cyclodextrin complexes

Shenyu Yang[1], Yun Yan[2], Jianbin Huang[2], Andrei V. Petukhov[3,4], Loes M.J. Kroon-Batenburg[5], Markus Drechsler[6], Chengcheng Zhou[2], Mei Tu[1], Steve Granick[7,8,9] & Lingxiang Jiang[1]

Proteins can readily assemble into rigid, crystalline and functional structures such as viral capsids and bacterial compartments. Despite ongoing advances, it is still a fundamental challenge to design and synthesize protein-mimetic molecules to form crystalline structures. Here we report the lattice self-assembly of cyclodextrin complexes into a variety of capsid-like structures such as lamellae, helical tubes and hollow rhombic dodecahedra. The dodecahedral morphology has not hitherto been observed in self-assembly systems. The tubes can spontaneously encapsulate colloidal particles and liposomes. The dodecahedra and tubes are respectively comparable to and much larger than the largest known virus. In particular, the resemblance to protein assemblies is not limited to morphology but extends to structural rigidity and crystallinity—a well-defined, 2D rhombic lattice of molecular arrangement is strikingly universal for all the observed structures. We propose a simple design rule for the current lattice self-assembly, potentially opening doors for new protein-mimetic materials.

[1] College of Chemistry and Materials Science, Jinan University, Guangzhou 510632, China. [2] Beijing National Laboratory for Molecular Sciences, State Key Laboratory for Structural Chemistry of Unstable and Stable Species, College of Chemistry and Molecular Engineering, Peking University, Beijing 100871, China. [3] Van't Hoff Laboratory for Physical and Colloid Chemistry, Debye Institute for Nano Materials Science, Utrecht University, Padualaan 8, 3584 CH Utrecht, The Netherlands. [4] Laboratory of Physical Chemistry, Department of Chemical Engineering and Chemistry, Eindhoven University of Technology, 5600 MB Eindhoven, The Netherlands. [5] Crystal and Structural Chemistry, Bijvoet Center for Biomolecular Research, Faculty of Science, Utrecht University, Padualaan 8, 3584 CH Utrecht, The Netherlands. [6] Bavarian Polymer Institute (BPI)—Laboratory for Soft-Matter Electron Microscopy, University of Bayreuth, D-95440 Bayreuth, Germany. [7] Center for Soft and Living Matter, Institute for Basic Science (IBS), Ulsan 44919, Republic of Korea. [8] Department of Chemistry, UNIST, Ulsan 44919, Republic of Korea. [9] Department of Physics, UNIST, Ulsan 44919, Republic of Korea. Correspondence and requests for materials should be addressed to L.J. (email: jianglx@jnu.edu.cn).

All living organisms are self-assembled entities where two major kinds of self-assembly are involved: the assembly of lipids into soft, fluidic membranes mainly driven by the hydrophobic interaction and the assembly of proteins into rigid, crystalline structures driven by a combination of hydrophobic, H-bonding and electrostatic interactions[1–12]. The lipid assembly is extensively reproduced and well extended by synthetic amphiphilic small molecules, polymers and even nanoparticles to form lamellar, tubular, vesicular and micellar structures (Fig. 1a)[13–15]. The protein assembly that produces crystalline structures (Fig. 1b)[2,3,9,10,16] such as lamellae, tubules and polyhedra is, however, largely unparalleled by synthetic or non-peptide molecules with a few notable exceptions[4,5,7,17–19]. The imbalance in the research of lipid and protein mimicry thus calls for more attention to the latter.

Looking beyond biomimetic self-assembly, one can notice that carbon allotropes[20] share the same morphological pattern: graphite, graphene, nanotubes and $C_{60}$ in analogy to lamellar, tubular and polyhedral assemblies. The properties and functions of carbon allotropes, lipid assemblies and protein assemblies are, however, drastically different as carbon atoms are connected by chemical bonds, lipids by hydrophobic interactions and proteins by a combination of intermolecular interactions. In this context, what has been gradually recognized is the importance of intermolecular interactions, structural flexibility/rigidity and fluidity/crystallinity over that of morphology. For example, peptoids[6], amphiphilic hexabenzocoronene[17] and catanionic surfactants[5,6,18,19] were found to form, respectively, nanosheets, nanotubes and regular hollow icosahedra, which successfully mimicked protein assemblies' morphologies but cannot rival their rigidity nor well-defined crystallinity.

Seeking for a synthetic system that parallels the morphology, rigidity and crystallinity of protein assemblies, we chose to study a supramolecular complex, sodium dodecyl sulphate (SDS)@2β-CD (one SDS molecule inside two β-cyclodextrin molecules, Fig. 1c). The complex was recently reported by us to form lamellar and tubular structures[21,22], but the study was restricted to morphology on the mesoscopic scale. In this paper, we reveal the existence of another structure, hollow rhombic dodecahedra and, more importantly, scrutinize the system on the microscopic/molecular scale to identify a high structural rigidity and well-defined, in-plane crystallinity.

## Results

**General phase behaviour.** Depending on concentration, the SDS@2β-CD aqueous solutions can be divided into lamellar (50 to 25 wt%), tubular (25 to 6 wt%) and polyhedral (6 to 4 wt%) phases (Fig. 1c). A general phase diagram and discussion on the SDS/β-cyclodextrin (β-CD) stoichiometry is given in Supplementary Fig. 1. The lamellar and tubular phases were studied in our previous work, so we focus here on the structural rigidity and newly identified nanosheets and polyhedra. Nanosheets can be found in the tubular and polyhedral phases as the minority and possibly the intermediate structure (Fig. 2a–c). The nanosheets are a few hundreds of nm in lateral size, very thin as reflected by the low electron contrast, and of a typical parallelogram shape with sharp, straight edges and an obtuse angle of 104°. When excess salts were added to the SDS@2β-CD solutions in all the 3 phases, flake crystals precipitated with a shape the same as the nanosheets but much larger and thicker (Fig. 2d). The microscopical image of the lamellar phase is full of parallel lines with a uniform interval (Fig. 2e), typical for cross sections of lamellar structures.

In the tubular phase, tubular structures pervades (Fig. 2f) with a monodisperse diameter ∼1 μm, a mean length ∼40 μm and open ends. A closer inspection reveals that the tube walls are

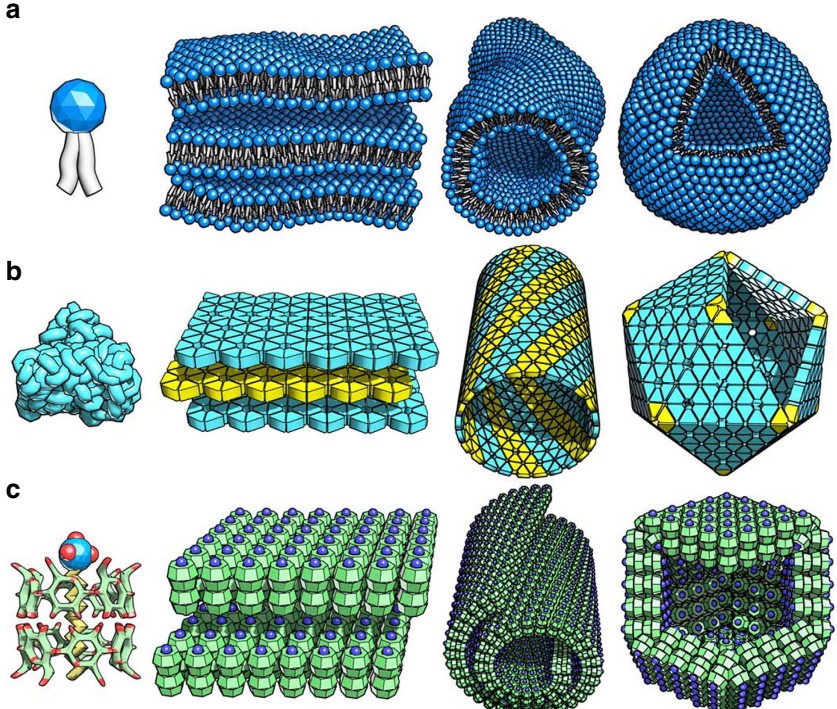

**Figure 1 | Stylized view of the lipid-like and protein-like self-assembly.** (**a**) Lipid molecules form lamellar, tubular and vesicular structures, the flexibility and fluidity of which are emphasized in the illustration. (**b**) Proteins form lamellar, helical tubular and regular icosahedral structures with rigidity and crystallinity (hexagonal lattice in this case). (**c**) SDS@2β-CD assembles, in a protein-mimetic way, into lamellar, helical tubular and rhombic dodecahedral structures with inherent rigidity and in-plane, rhombic crystalline nature. In the molecular view, SDS is a anionic surfactant with a hydrocarbon tail (yellow) and a -$(SO_4)^-$ headgroup (blue and red), while β-CD is a ring of hepta-saccharides (green C and red O atoms).

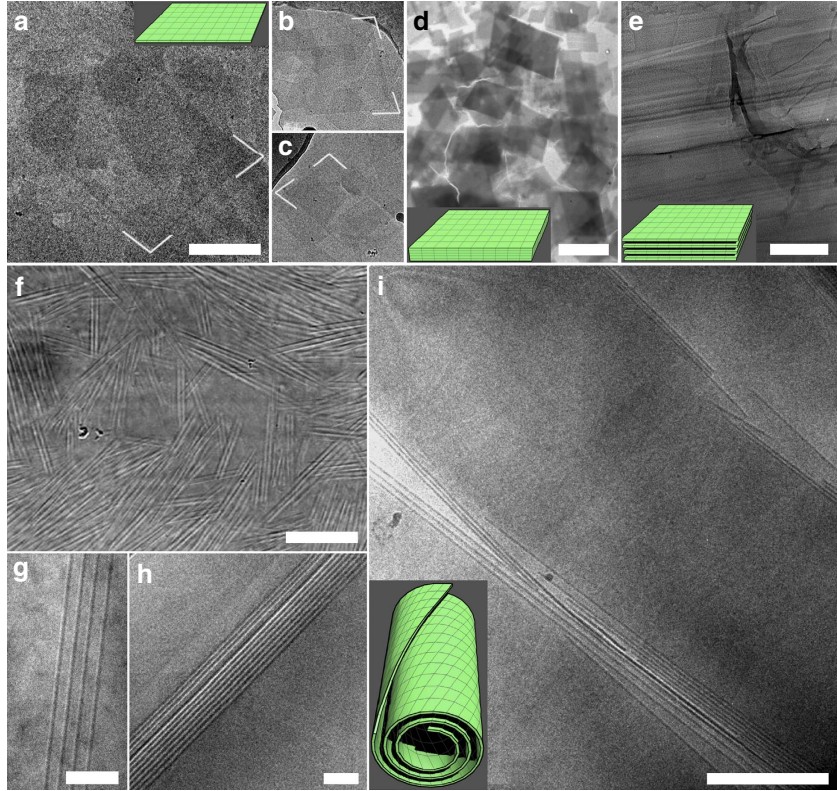

**Figure 2 | Morphologies of the nanosheet-based and tubular structures formed by SDS@2beta-CD complexes.** (**a–c**) Cryo-TEM pictures of parallelogram nanosheets with an obtuse angle of 104°, scale bar, 200 nm. (**d**) A TEM image of flake crystals with a shape identical to that of the nanosheets, scale bar, 5 μm. (**e**) A freeze-fractured TEM picture of lamellar structures, scale bar, 100 nm. (**f–i**) Pictures of multilamellar tubular structures: an overview (**f**, optical microscopy, scale bar, 20 μm), zoom-in of the tube walls (**g** and **h**, cryo-TEM, scale bar, 100 nm), and a fractured tube (**i**, cryo-TEM, scale bar, 500 nm).

made of multiple, equally spaced and $\sim 4$ nm thick (for detailed discussion about layer thickness, please see Supplementary Fig. 2) layers (Fig. 2g–i). In a situation like Fig. 2i, the walls are fractured rather than bended. The straightness of the tubes is strikingly persistent across at least 4 orders of length scale: no mesocopical bending on the order of 100 μm (Fig. 2f) nor microscopical fluctuation on the order of 10 nm (Fig. 2g,h). The persistence length of the tubes is estimated to be at least on the order of 1 m (Supplementary Fig. 3), order-of-magnitude higher than that of lipid tubules $\sim 10$ μm (ref. 23) and microtubules $\sim 1$ mm (ref. 24), signifying a high rigidity.

**Hollow rhombic dodecahedra**. In the polyhedral phase, transmission electron microscopy (TEM) pictures show polygons, mostly hexagons and some octagons, $\sim 1$ μm large with sharp edges (Fig. 3a,d). We found that the cross sections of a rhombic dodecahedron can account for the observed polygons. In Fig. 3c, a rhombic dodecahedron is rotated along $x$ and $y$ axis and its cross sections along the $xy$ plane are represented by the dark grey polygons. The indexed polygons are compared with the TEM pictures in Fig. 3d, reaching a good agreement. The freeze-fractured (FF) TEM pictures (Fig. 3d, middle row) reveal that the dodecahedra are hollow and, sometimes, of double shells. The cryogenic transmission electronic microscope (cryo-TEM) pictures (Fig. 3d, bottom row) show a normal dodecahedra and a rare one with a nonconcentric dodecahedron inside.

The rhombic dodecahedral geometry is further consolidated by the atomic force microscopy (AFM) measurements (Fig. 3b), in which bumpy, rhombic objects can be observed. We reason that the observed morphology is resulted by the collapse of hollow dodecahedra on a flat substrate in dry condition (Fig. 3e, see the

figure caption). In the case of a single-shell dodecahedron (shell thickness 4 nm), the bumps, sink and pits are expected to be higher than, close to, and lower than 8 nm, respectively. These expectations are confirmed by two examples in Fig. 3f, where the yellow-white, green and green-blue correspond to heights of 16 nm, 8 and 4 nm, respectively. We therefore determined that the structures are of a hollow rhombic dodecahedral geometry, which has not been observed before in self-assembly systems and is of potential uses in templated particle synthesis and colloidal self-assembly.

**Universal in-plane molecular arrangement—a 2D rhombic lattice**. X-ray scattering technique was employed to resolve the molecular arrangement of SDS@2β-CD inside the observed structures. Small-angle X-ray scattering (SAXS) measurements of the lamellar, tubular and polyhedral phases (microstructures randomly oriented) were conducted in European Synchrotron Radiation Facility, Grenoble, France. The SAXS curves can be divided into 3 regions (Fig. 4a). In the cyan region ($0–4$ nm$^{-1}$), 4 curves share a similar decaying trend corresponding to the form factor. The pink region features equally spaced, lamellar peaks that gradually vanishes on the dilution of SDS@2β-CD, indicating the increase of layer-to-layer distance (from 15 to 42 nm) and the lessening of layers in a stack. This is in line with the TEM observations where the lamellar, tubular and polyhedral structures have a dozen of, several and one or two coherent layers, respectively (Figs 2 and 3). In the green region ($4–8$ nm$^{-1}$), 3 distinctive structural peaks can be identified for all the curves, implying a single in-plane crystalline arrangement shared by all the microstructures. The scattering curves up to 4 nm$^{-1}$ were fitted by a bilayer form factor and lamellar structure factor (modified Caillé theory[25], Fig. 4b). The corresponding electron density profile suggests a bilayer

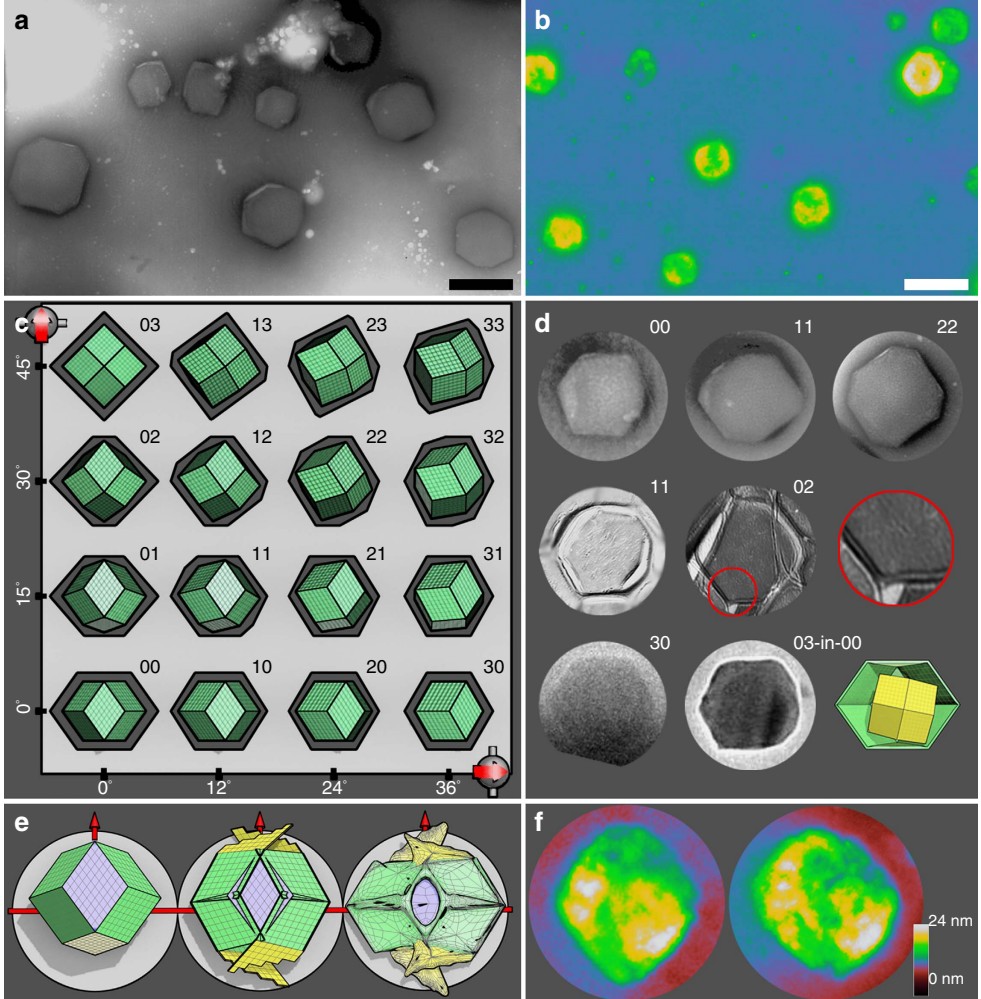

**Figure 3 | Morphology of the SDS@2β-CD polyhedra.** (**a**) A TEM picture of the negatively stained polyhedra, scale bar, 1 μm. (**b**) An AFM image of the polyhedra on mica substrate in a dry condition, scale bar, 2 μm. (**c**) A diagram showing the top view and cross-sectional view (along the *xy* plane, the dark grey polygons) of a rhombic dodecahedron with different orientations. The orientation is indexed to be compared with TEM images. (**d**) Selected TEM pictures corresponding to the rhombic dodecahedral geometry with varied orientations. From top to bottom rows are images by negatively stained TEM, FF-TEM (highlighting a double-shell configuration), and cryo-TEM (featuring a rare, nonconcentric configuration). (**e**) A simple scenario of the collapse of hollow dodecahedra. First, one of the 12 equal faces rather than any vertex will land on a flat substrate, then the 2 faces parallel to the substrate are labelled cyan, the 2 faces perpendicular to the substrate yellow and the other 8 faces green. Second, we consider the dodecahedron to experience a vertical pressure due to gravity and dehydration, then the cyan faces may remain intact, the yellow faces may be forced to fracture, and the green faces squeezed. Finally, the dodecahedron collapses into a somewhat rhombic structure with bumps along the equator, a sink in the center, and pits near two poles. (**f**) Selected AFM images are accounted by the scenario. Please note that the dodecahedra are about 1 μm large so the scale bars for **d** and **f** are not shown.

arrangement of SDS@2β-CD with its long axis perpendicular to the layer (Fig. 4c). In Caillé theory, the Caillé parameter $\eta$ describes the thermo fluctuations of the layers, inversely related to the layer rigidity. The fitted $\eta$ (on the order of 0.01) underlines that the rigidity of SDS@2β-CD microstructures is order-of-magnitude higher than that of lipid membranes ($\eta$ usually on the order of 1)[25].

To precisely determine the in-plane lattice structure, we aligned the tubular structures and then studied the sample with wide-angel X-ray scattering. By carefully loading tubular samples into capillaries, we achieved flow-induced alignment of tubes (Supplementary Fig. 4). The scenario of a multilamellar tube diffracting X-ray is schematically illustrated in Fig. 4d, where the lamellar period (10s of nm) gives cyan dots at small q along the equator and the in-plane lattice (about the molecular size, ∼1 nm) yellow dots at large q. In line with this scenario, a few lamellar peaks and a broad form factor peak are located on the equator at small q (Fig. 4e), while a well-defined diffraction pattern is observed at large q (Fig. 4f). The pattern's symmetry along the equator and

medial axis signifies a helical nature for the tubes. The pattern is perfectly matched by two mirrored lattices (green and yellow grids).

The in-plane unit cell is thus resolved as a rhombus with $a = b = 1.52$ nm (comparable to β-CD diameter, 1.5 nm) and $\gamma = 104°$, in excellent agreement with the reported single-crystal data for other aliphatic chain/β-CD complexes[26]. With the $b$ axis-to-equator angel = 3°, the lattice scroll up right-handed or left-handed into helical tubes (Fig. 4g). Such a lattice is stabilized by an extensive network of direct (CD to CD) H-bonds and indirect (water mediated) H-bonds[26], where the O atoms on two neighbour CD rims are close enough to form multiple direct H-bonds (highlighted by magenta in Fig. 4h, the top and side views). In addition, we argue that the $\gamma$ 104° is a consequence of maximization of the direct H-bonds with the given 7-fold molecular symmetry of β-CD (Supplementary Fig. 5).

It is noteworthy that the rhombic lattice is universally presented in all the observed microstructures as evidenced by the identical in-plane structural peaks in all the 3 phases (Fig. 4a,

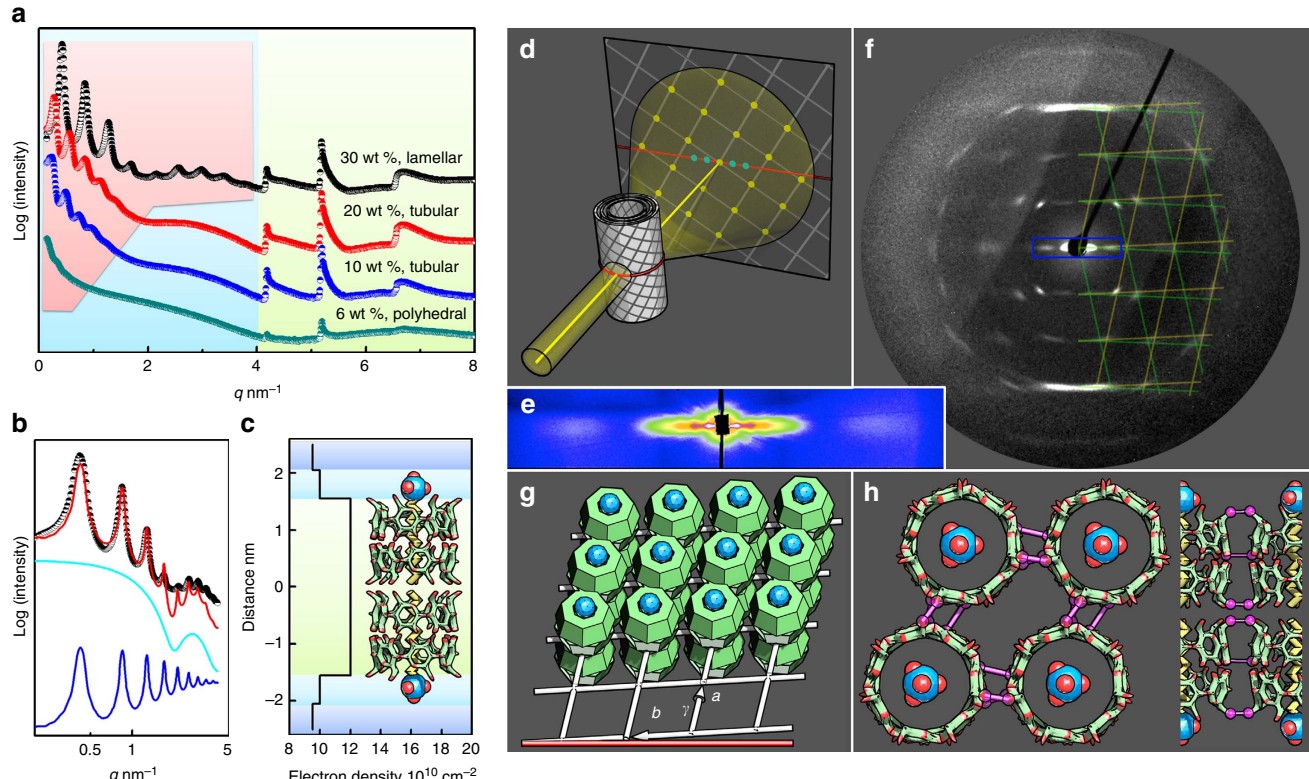

**Figure 4 | X-ray scattering of the SDS@2β-CD structures. (a)** SAXS curves of the lamellar, tubular and polyhedral phases with different SDS@2β-CD concentrations. The curves (vertically shifted for clarity) are divided into pink, cyan (overlapping with pink) and green regions, corresponding to lamellar structure factor, form factor and in-plane crystallinity, respectively. **(b)** The SAXS curve (30 wt%, black dots) up to 4 nm$^{-1}$ is fitted by a bilayer form factor (cyan line) and lamellar structure factor (blue line) with a good result (red line). The cyan and blue lines are vertically shifted for clarity. **(c)** The electron density profile across the bilayer. The thickness 3.1 nm and electron density of the core part (the green region) matches a vertical stack of 4 β-CD molecules (0.75 nm high), the thickness and electron density of the outer part suggests two SDS molecules with their tails threading the CD cavities and headgroups pointing outwards. Threaded β-CD molecules usually stack in a head-to-head fashion to maximize the inter-CD H-bonding along the threading direction. **(d)** A scheme of wide-angel X-ray scattering of aligned multilamellar tubes with in-plane crystallinity. **(e)** Scattering pattern along the equator at small angel, corresponding to the blue box in **f**. **(f)** Scattering pattern at large angel is super-positioned by the mirrored yellow and green grids. **(g)** The in-plane lattice with a rhombic unit cell, $a = b = 1.52$ nm, $\gamma = 104°$ and the $b$ axis-to-equator angel = 3°. **(h)** Top and side views of the SDS@2β-CD lattice, highlighting the possible, inter-CD H-bonds (magenta).

green region) and by the magic angle 104° appeared for the nanosheets (Fig. 2a–c), precipitated flakes (Fig. 2d) and tubes (Fig. 4f). We therefore termed the assembly of SDS@2β-CD as lattice self-assembly, the resultant structures of which feature a well-defined in-plane crystalline nature. The nanosheets serve as building blocks or intermediate structures to laterally expand into lamellae, to scroll up helically into multilamellar tubes, to fold into hollow rhombic dodecahedra, and to stack up into flake precipitates depending on the concentration of SDS@2β-CD and salts (Fig. 1c). In contrast to the traditional self-assembly of lipids or amphiphilic molecules that are fluidic and soft, the lattice self-assembly of SDS@2β-CD is crystalline and rigid, resembling the protein assembly such as viral capsids. Although the SDS@2β-CD tubes and polyhedra and viral capsids are similar in shape, the formers are much larger in size.

It is interesting to note that a synthetic octapeptide, lanreotide, was reported to form monodisperse nanotubes (although not lamellar nor polyhedral structures)[27]. The observed lanreotide bilayers are of some resemblance to the present 2D rhombic lattice, where the former is of well-defined, in-plane helical crystallinity and the hydrophobic residues are protected from water by the inner and outer β-sheet fibres and by the hydrophilic residues.

### Possible pores at the vertices of the rhombic dodecahedra. We noticed in the EM pictures that the dodecahedra vertices are usually

rounded, indicative of pores at the vertices. Following Dubois and Zemb *et al.*'s arguments[4,5], we reason two possible ways to understand the formation of pores. One is to remove the highly undesired curvature singularity at the vertices by forming pores. Another is to minimize total curvature energy by creating negative Gaussian curvature at the pores and leaving the spontaneous curvature and Gaussian curvature to be close to zero at the facets. The zero curvature is clearly favoured as evidenced from the predominant lamellar and tubular (nearly planar due to the large diameter) structures at higher concentrations. The negative local curvature could be partially relieved by higher surface charge at the pores.

In the previous catanionic icosahedral system[4,5], the excess anionic surfactant molecules enrich themselves at the pores to form half micelles to cover edges of the pores. In the current case, if the hydrophilic outer rims of CD are exposed to water at the edges of pores or planar structures, its energy penalty is much less than that of exposing hydrophobic chains to water in the catanionic surfactant system. It is, therefore, not clear at this stage about the exact molecular arrangement at the pores. For extended discussion on the rhombic dodecahedral geometry, please see the Supplementary Discussion[28,29].

### Encapsulation by the SDS@2β-CD tubes. The main function of viral capsids and bacterial compartments is to encapsulate DNA, RNA, proteins and even magnets[2,9,16]. In this context, we recently

**Figure 5 | Applications of the lattice self-assembly of SDS@2β-CD.** (**a**) A confocal laser scanning microscopy image of monodisperse 800 nm liposomes inside the tubes, scale bar, 4 μm. (**b–d**) Schematic representations of the CD networks formed after guest removal. Hexagonal (**b**), rhombic (**c**) and square (**d**) lattices of α-CD, β-CD and γ-CD, respectively. The cyan and yellow regions indicate hydrophilic and hydrophobic pores, respectively.

reported that the SDS@2β-CD tubes can spontaneously enclose colloidal particles with different shapes and chemistry to form 1D straight chains, double/triple helices and hybrid structures[30,31]. An interesting case is the haematite chain inside a tube similar to the magnetosome chain inside magnetotatic bacteria[9]. In this work, we take a step forward by including soft liposomes into the tubes. Since SDS is a surfactant capable of solubilizing lipids, liposomes with a high cholesterol content were chosen to resist SDS[32]. The added liposomes ∼0.8 μm remain intact for at least several hours and are indeed trapped in the tubes (Fig. 5a). This hierarchical, liposome-in-tube construction reminds us of bacterial walls where the lipid membrane are covered by a surface layer of crystalline proteins (S-layer)[10]. It is envisioned that viable bacteria (typically ∼1 μm large) could be included into the tubes, enabling the research of bacterial confinement and assembly.

## Discussion

The generalization of the lattice self-assembly is established in a matrix of CD complexes with the guest molecules ranging from ionic, zwitterionic, to nonionic amphiphilic molecules and the host molecules being α-CD and β-CD. Replacing β-CD with 2-hydroxypropyl-β-CD or methyl-β-CD makes the complexes fully dissolvable in water and unable to assemble, underlining the necessity of inter-CD H-bonds (see also the discussion of Fig. 4h and Supplementary Fig. 5 for the importance of the H-bonding network). Recalling that excess of salt results in precipitation, we argue a simple design rule for the lattice self-assembly: the presence of strong, directional in-plane attraction (H-bonds in this case) and out-of-plane repulsion (for example, electrostatic or steric). The insensitivity of the assembly behaviour to the head group of the guest molecules suggests that one can functionalize the head group to mimic bacterial compartments for bio-mineralization and enzyme catalysis[9,16]. Different host molecules are expected to produce fairly different lattices: hexagonal for α-CD, rhombic for β-CD and square for γ-CD depending on the molecular symmetry (6-, 7- and 8-folds, respectively, Fig. 5b). Looking ahead to possible applications, fixation of the CD networks by polymerization and subsequent removal of the guest molecules could form selective membranes as well as porous tubes and polyhedra with hydrophobic channels inside CD molecules and hydrophilic channels between CD molecules (Fig. 5c, the yellow and cyan regions). The pore geometry might be precisely tuned by employing different CD molecules. Such membranes are, in principle, similar to the reported colloidal kagome lattice[33] but with much smaller pore sizes.

## Methods

**Materials.** SDS (99%) was purchased from Acros Organics. and used as received. β-CD was purchased from Sinopharm Chemical Reagent with a water content of 14%. The silica particles with polydispersity <5% and haematite particles with polydispersity <10% were synthesized in the van't Hoff Laboratory for Physical and Colloid Chemistry in Utrecht University. 1-Palmitoyl-2-oleoyl-sn-glycero-3-phosphocholine and cholesterol (98%) were obtained from Avanti Polar Lipids Inc. β-BODIPY 530/550 C$_5$-HPC was purchased from Molecular Probes. HEPES (sodium salt) was obtained from Acros Organics.

**Preparation of SDS@2β-CD suspensions.** Desired amounts of β-CD, SDS and water were weighed into tubes to give a constant β-CD/SDS molar ratio of 2:1 and different total concentrations of β-CD and SDS (in wt%). The mixtures were heated to ∼60 °C to obtain transparent and isotropic solutions, where SDS@2β-CD is formed. The solutions were then cooled to 25 °C and incubated at least 48 h to allow SDS@2β-CD to form structures. According to the morphology of the structures, the whole studied concentration range is divided into three phases, lamellar (25–50 wt%, semi-transparent and highly viscous), tubular (6–25 wt%, turbid and viscous) and polyhedral (4–6 wt%, slightly turbid and watery), respectively. The structures therein are stable for months.

**Liposome preparation.** A composition of 1-palmitoyl-2-oleoyl-sn-glycero-3-phosphocholine/cholesterol (0.63/0.37, molar ratio) was chosen to resist SDS solubilization. These constituents were mixed using chloroform and dried overnight under vacuum to give lipid films in glass vials. HEPES buffer was used to hydrate these films to form multilamellar vesicles. After three freeze-thaw cycles, this solution was extruded using a 1 μm pore size membrane in a miniextruder by Avanti Polar Lipids, producing monodisperse, unilamellar vesicles ∼0.8 μm in diameter.

**Co-assembly of colloidal particles or liposomes with SDS@2β-CD tubes.** Aqueous suspensions of colloids were centrifuged at 2,000–3,000 r.p.m. for 15 min, followed by removal of the supernatant water. A 10 wt% SDS@2β-CD tubular suspension was added to the centrifuge tube to give a final tube/colloid mixture containing 1–20 wt% colloids. The sample was heated to ∼60 °C to melt the tubes and was sonicated to disperse the particles. Then the centrifuge tube was cooled to room temperature and incubated with gentle vibration to avoid sedimentation of the particles. On cooling, the tubes formed and the particles were spontaneously included into the tubes.

**Transmission electron microscopy.** TEM images were recorded on a JEM-100 CX II transmission electron microscope (JEOL, Japan, 80 kV). The samples were prepared by dropping solutions onto copper grids coated with Formvar film. Excess water was removed by filter paper, and the samples were dried in ambient environment at room temperature for TEM observation. In case of the polyhedral samples, they were negatively stained by uranyl acetate before water removal.

**Freeze-fracture transmission electron microscopy.** A small drop of a sample was placed between two copper disks. The mounted sample was plugged into liquid propane that was cooled by liquid nitrogen. Microstructures are preserved by this procedure most of cases. A freeze-fracture apparatus (BalzersBAF400, Germany) was employed to fracture and replicate the specimen at −140 °C. A thin layer of platinum-carbon was cast onto the specimen to obtain the replicas, which were then ready for TEM observation.

**Cryogenic transmission electronic microscope.** A few microlitres of samples were mounted onto a copper TEM grid and the excess solution was removed with filter paper. The specimen was immersed into a cryo-box (Carl Zeiss NTS GmbH, filled with liquid ethane) to be rapidly cooled to −170 to −180 °C. The specimen was then transferred to a Zeiss EM922 EFTEM (Zeiss NTS GmbH, Oberkochen, Germany) by a cryo-transfer holder (CT3500, Gatan, Munich, Germany). TEM observation was made at −180 °C with an acceleration voltage of 200 kV. Reduced electron doses (500–2,000 e nm$^{-2}$) were used to obtain zero-loss filtered images.

A bottom-mounted CCD camera system (UltraScan 1,000, Gatan) was employed to record all images, which were then processed by Digital Micrograph 3.9.

**Confocal laser scanning microscopy.** The SDS@2β-CD structures were stained by Nile red in the following way. A drop of 15 μl Nile red in acetone solution (1 mg ml$^{-1}$) was added to a test tube, followed by volatilization of the acetone. A desired amount of SDS@2β-CD aqueous solution was then added to the tube. The samples were then incubated at least 24 h for the dye molecules to diffuse into the SDS@2β-CD structures. The stained SDS@2β-CD, colloid-in-tube, or liposome-in-tube suspensions were loaded into glass capillaries (Vitrocom, 0.1 × 2 × 50 mm), which were then sealed by a ultraviolet-curing epoxy glue. An inverted confocal laser scanning microscopy (Leica, True Confocal Scanner SP, Germany) was used to conduct experiments in florescence and differential interference contrast modes.

**Atomic force microscopy.** AFM measurements were conducted by Nanoscope IIIa (Digital Instruments Inc., USA) in tapping mode under ambient conditions on mica substrates.

**Small-angle X-ray scattering.** Measurements were performed at the DUBBLE beamline of the European Synchrotron Radiation Facility (ESRF, Grenoble, France). The microradian resolution setup was used. A PILATUS-1M detector (consisting of 18 multi-chip modules covering a total area of 243 by 210 mm$^2$) was placed at a distance of 3.4 m from the sample. The selected wavelength of the X-rays was 0.095 nm, the beam size at the sample position about 0.3 mm. The intensity profiles (Fig. 4a) were obtained by integrating over a circular sector containing the area of interest in the scattering patterns. The scattering curves were fitted by SASfit software package[34].

**Data availability.** The authors declare that the data supporting the findings of this study are available within the article and its Supplementary Information Files.

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

## Acknowledgements

We appreciate the insightful suggestions from Thomas Zemb. L.J. acknowledges support by the startup funding from Jinan University. L.J. thanks helpful discussion with Albert P. Philipse and Samia Ouhajji. S.G. acknowledges support by the Institute for Basic Science, project code IBS-R020-D1. The personnel of the DUBBLE beamline is thanked for the support during SAXS measurements. The Nederlandse Organisatie voor Wetenschappelijk Onderzoek (NWO) is acknowledged for the provided synchrotron beam-time.

## Author contributions

L.J. conceived the experiment and wrote the manuscript; S.Y performed the experiments. L.J., Y.Y. and J.H. discussed microscopy results; L.J., A.V.P. and L.M.J.K.-B. performed the X-ray measurements and discussed the results; M.D. took the cryo-TEM pictures. All authors contributed to analysing the results and writing the paper.

## Additional information

**Competing interests:** The authors declare no competing financial interests.

