## [Peer Review File · Nature Communications]

Reviewers' comments:

Reviewer #1 (Remarks to the Author):

The results are exciting and the system well characterized. I recommend its publication. The authors find a series of morphologies that resemble biological assemblies using a supramolecular molecule. I found fascinating that the closed structure has a rhombic crystal in plane. This is not expected because the defects required to close a crystalline structure with lower symmetry of hexagonal in 2D required having defects that are not very prominent (8 defects with 3 nearest neighbors or triangles). I would like the authors to comment on this before the paper is published. In the icosahedra there are 12 defects with 5 nearest neighbors (pentagons) that cause strain in the hexagonal lattice and so they are further apart from each in the position of the 12 vertices of the icosahedra. One would expect in a lattice with 4 nearest neighbors the defects would be 8 triangles defects that would simultaneously buckle into a cube with eight vertices. A way out is having large pores that change the topology of the closed surface and in that case the topological constraints change regarding the number of defects making possible the buckling into structures other than the cube, such as the rhombic dodecahedra found here but that is not the case at least in their Figure 1. I am not sure what is the buckling mechanism into rhombic dodecahedra. In Figure 1 the authors made a sketch of the closed structure and it seems that in the vertices they have defects (triangles?). How many defects they needed to draw such structure? In principle the rhombic dodecahedra has 14 vertices: Are there 14 defects added in the figure to close the planar rhombic crystal into a sphere? If so what type of defects were selected (triangles I assume)? There are very solid theorems that would restrict the number of defects in close lattices (Euler theorem) and it would be truly terrific if the authors can explain or at least illuminate the reader about how a closed rhombic lattice buckles into a rhombic dodecahedra (which dual is a cuboctahedron, an Archimedean solid, and not a Platonic solid made of equal polygons but as an Archimedean solid though made of 2 type of polygons, squares and triangles, all vertices should be equivalent). I guess buckling into platonic solid which faces are made by identical polygons are simpler to explain. I find the hollow crystalline shells found here truly fascinating since the faces have to be of 2 types of polygons so they break strong symmetry arguments. I would like to comment that this is the first time I have seen buckling of a rhombic 2D crystal in a closed surface into a hollow rhombic dodecahedra. If it is possible to characterize the defects that would be a very attractive result that will lead to more fundamental studies.

Reviewer #2 (Remarks to the Author):

The work presented by Granick and co-workers clearly extends concepts first developed for lipids, later for peptides, now to inclusion complexes made of a cyclodextrin and a targeted filling electrolyte. Since the SDS is charged, these type of hybrid colloids are governed by electrostatic effects, as well as by packing given by the shape of the assembled elementary bricks. This goes in the direction of the pioneering work presented by Granick in his classical paper in Science (2011) – modestly not cited, but goes a step significantly further.

The key arguments making this deserving of publication in Nature are the following :

1/ figure 1 introduces a new general concept, easily understandable without background knowledge in statistical phys. chem.

2/ The different examples given in real space (EM) are easy to follow, as well as the really multi-scale scattering shown in reciprocal space.

3/ the statistics in number of edges and tiling of projected objects gives another example of the importance of edge and vertex counting: free energy of formation is driven by the competition between solid angle along edges and vertex free energy excess.

These three elements make this paper very likely to become a highly cited "classic".

Careful reading about methods of observation and sample preparation don't give me any point that seems not clear and carefully established. In other case of solid self-assembly where the building block is crystalline, the importance of stoichiometry is crucial: being off-stoichiometry would change the morphology of the aggregate as well as the average average size of objects. How can the authors be sure that all (>99%) of the cyclodextrins are loaded with SDS? What happens if SDS is in excess or cyclodextrin cages are in excess? This would require to add of a paragraph (or longer in Suppl. Materials). Reference 20 and 21 of the same research group deal with this point in general, but not enough so the association-dissociation equilibrium and salinity dependence can be understood for the samples described in this work;

Close examination of the text and arguments given show to me a certain number of weak points, mainly connected to similar study of self-assembly at colloidal scale.

-figure 1 b and c are close to the four levels of organisation as observed for lanreotides with three subunits, similar to the outside rim, the charged and the uncharged face of cyclodextrin (PNAS 2003). The analogy should be noted and could be commented. For the spiralled shapes in the figure 1c, the general theory of Aggeli and Boden should apply, in the case of vanishing torsion angle of the elementary building block. This should be explained here (see figure 1 of Aggeli, A in PNAS 2001

- the theory as "assembly by a child" in Kaspar-Klug mechanisms, are definitively excluded in the result of self assembly of surfactant-loaded cyclodextrin. It seems to me that the evaluation of edge and vertex free energy, as detailed in the figure 5 of Dubois, M PNAS 2004, applies. This could be confirmed (or dismissed the corresponding paragraph of text (page 4 of this manuscript).

-

Finally, there seem to be an extensive literature about buckling transition that is necessary to switch between faceted and bent structures as shown here, the two more known by David Nelson and Antonio Silber. Due to the expected impact of the present work, it could be useful for the reader to know if theoretical expectations are met (or not) in the experimental example shown in this work.

Thomas Zemb

Response to Reviewer # 1

We thank the reviewer, who appreciated the novelty of the observed hollow rhombic dodecahedral structure and recommended publication of the manuscript. We have done our best, in the revised manuscript, to address the reviewer's concerns.

Concern about defects at the vertices of the rhombic dodecahedra.

We realized that this question is indeed central to polygonal geometries but was not sufficiently discussed in the previous submission. Our thoughts in the revised manuscript are the following.

a) Assume, for the moment, that the rhombic lattice is ideally 2D (infinitely thin). Then the lattice can form fully close shell of rhombic dodecahedra with two kinds of vertices: threefold and fourfold ones with 3 and 4 adjacent faces, respectively. The threefold vertices have 3 nearest neighbors, while the fourfold vertices have 4 nearest neighbors.

b) However, we incline to the formation of holes at the vertices. We noticed in the EM pictures that the dodecahedra vertices are usually rounded, indicative of pores at the vertices. Following Dubois and Zemb et. al.'s arguments (Dubois, M., B. Deme, et al. (2001). "Self-assembly of regular hollow icosahedra in salt-free catanionic solutions." *Nature* 411(6838): 672-675.), we reason two possible ways to understand the formation of pores. One is to remove the highly undesired curvature singularity at the vertices by forming pores. Another is to minimize total curvature energy by creating negative Gaussian curvature at the pores and leaving the spontaneous curvature and Gaussian curvature to be close to zero at the faces. The zero curvature is clearly favoured as evidenced from the predominant lamellar and tubular (nearly planar due to the large diameter) structures at higher concentrations. The negative local curvature could be partially relieved by high surface charge at the pores.

In the previous catanionic icosahedral system, the excess anionic surfactant molecules enrich themselves at the pores to form "half" micelles to cover edges of the pores. In the current case, the hydrophilic out rims of CD are exposed to water at the edges of pores or planar structures, so its energy penalty is much less than that of exposing hydrophobic chains to water in the catanionic surfactant system. It is, therefore, not clear at the present about the exact molecular arrangement at the pores.

It is pleasing to see the present self-assembly of hollow rhombic dodecahedra as a manifestation of the quasi-equivalence principle—the association of identical building

units into highly symmetric structures due to strong intermolecular H-bonding with high efficiency and no templates. It is also surprising that, unlike the icosahedral geometry favoured by cationic surfactants and many capsid proteins, the rhombic dodecahedral geometry is dominant possibly as a consequence of the in-plane rhombic lattice.

The above discussion is added to the revised manuscript under the subtitle “Possible pores at the vertices of the rhombic dodecahedra”. Further quantitative calculation of free energies of vertices and edges is difficult at the current stage due to the limited knowledge of the exact molecular arrangement at the pores, the bending energy, and other quantities. We plan to study this particular subject in a follow-up project.

Concern about how a planar rhombic lattice buckled into a rhombic dodecahedron.

The current dodecahedra were formed by CD complexes (the building units) at a desired concentration via reversible self-assembly: at high temperature ~ 60 C the building units are fully dissolved in water, forming no dodecahedra nor planar lattice, at room temperature the building units assemble into dodecahedra (up to 1 micron) in coexistence with a minority of planar structures (a few hundreds of nm). There are three possible pathways of dodecahedron formation: 1) the building units add in one by one to form the dodecahedra, 2) small planar structures (several hundreds of nm, like what we observed) assemble into the dodecahedra, and 3) a single large planar structure (has to be several micron large) buckles into a single rhombic dodecahedron. As we did not observe any planar structure that large, we speculate that the buckling pathway is of lower possibility and that the former two pathways or their combination are of higher possibility. This paragraph is added into the supplementary information of the revised paper. Of course, the general buckling mechanism that goes beyond the current manuscript is indeed a fundamental subject worthy studying.

Extra discussion on this matter is also given in the end of supplementary information of the revised manuscript.

Concern about two kinds of polygons in the faces of rhombic dodecahedra.

As shown in the above figures, the rhombic dodecahedron has one kind of faces, thus one kind of polygons and lattices, but does have two kinds of vertices.

Response to Reviewer # 2

We thank the reviewer, Thomas Zemb, for his insightful suggestions and for his recommendation of publication. Zemb and coworkers' pioneering work on the regular icosahedra, discs, and punctuated planes formed by cat-anionic surfactants was indeed an inspiration to the present work in many aspects. We have done our best, in the revised manuscript, to address the reviewer's concerns.

Concern about the stoichiometry of SDS and CD and the general phase behavior.

As the reviewer suggested, the stoichiometry is indeed crucial to crystallinity. We previously did some experiments relevant to this aspect and add the results to the supplementary information of the revised paper.

A general phase diagram is shown below (Figure S1). Along the SDS/ β -CD ratio axis, plate crystal precipitates tend to form at low ratio because β -CD is of very limited solubility in water, and the solution is clear without any observable aggregates at high ratio because SDS/ β -CD 1/1 complex is of excellent solubility in water. Near the 1/2 stoichiometry line, the complexes form lamellar, tubular, and polyhedral structures depending on the concentration. Although the current paper is focused on the assembly behavior along the stoichiometry line, we did measure the SDS/ β -CD ratio inside assembly structures using PGSE-NMR method for a few selected samples that are a bit off the stoichiometry line (crosses in the Figure S1). Briefly, this method can measure the concentrations of free SDS and β -CD molecules that do not participate into the assembly structures because free molecules are of high diffusivity. For details of this method, please see our previous publication (Lingxiang Jiang et. al. Selectivity and Stoichiometry Boosting of β -Cyclodextrin in Cationic/Anionic Surfactant Systems: When Host–Guest Equilibrium Meets Biased Aggregation Equilibrium J. Phys. Chem. B. 2010, 114, 2165–2174). According the results, we conclude that the SDS/ β -CD ratio inside the assembly structures is always very close to 1/2 (at least 98% of the complexes are in the 1/2 form) even when the bulk ratio is a bit off 1/2.

Figure S1. The general phase diagram of the SDS/ β -CD aqueous solution at room

temperature.

Relevance of Fig. 1b and c to previous work (PNAS 2003 and PNAS 2001).

We thank the review for noticing the relevance to lanreotide self-assembly and the general theory of Aggeli, A et al.

It is interesting to note that a synthetic octapeptide, lanreotide, was reported to form monodisperse nanotubes (although not lamellar nor polyhedral structures). The observed lanreotide bilayers are of some resemblance to the present 2D rhombic lattice, where the former is of well-defined, in-plane helical crystallinity and the hydrophobic residues are protected from water by the inner and the outer β -sheet fibers and by the hydrophilic residues. These sentences are added to the revised paper,

The theory of A. Aggeli et al. was developed for β -sheeting forming peptides that assemble into twisted tapes and ribbons, helical cylinders, and fibers, where the building units were modelled as chiral rods with donor and acceptor groups. The current cyclodextrin complex is indeed rod-like, but its chirality and donor-acceptor groups are not immediately quantifiable. The twist tapes (the basic structure in Aggeli theory) is not observed in our work. And the tube diameter is so large (~ 1 micron) that the torsion and twist of cyclodextrin complexes are negligible in our case. So we tend to make the connection when more evidences emerge.

Concern about the vertices and edges of the rhombic dodecahedra.

We agree that the PNAS 2004 paper are of high relevance to the current observation of polyhedral structures. The references are now properly cited in the revised manuscript along with 3 paragraphs of discussion on the relevance and especially on the pores.

We noticed in the EM pictures that the dodecahedra vertices are usually rounded, indicative of pores at the vertices. Following Dubois and Zemb et al.'s arguments, we reason two possible ways to understand the formation of pores. One is to remove the highly undesired curvature singularity at the vertices by forming pores. Another is to minimize total curvature energy by creating negative Gaussian curvature at the pores and leaving the spontaneous curvature and Gaussian curvature to be close to zero at the faces. The zero curvature is clearly favoured as evidenced from the predominant lamellar and tubular (nearly planar due to the large diameter) structures at higher concentrations. The negative local curvature could be partially relieved by high surface charge at the pores.

In the previous cationic icosahedral system, the excess anionic surfactant molecules enrich themselves at the pores to form "half" micelles to cover edges of the pores. In the current case, the hydrophilic out rims of CD are exposed to water at the edges of pores or planar structures, so its energy penalty is much less than that of exposing hydrophobic chains to water in the cationic surfactant system. It is, therefore, not clear at the present about the exact molecular arrangement at the pores.

The above discussion is added to the revised manuscript under the subtitle "Possible pores at the vertices of the rhombic dodecahedra". Further quantitative

calculation of free energies of vertices and edges is difficult at the current stage due to the limited knowledge of the exact molecular arrangement at the pores, the bending energy, and other quantities. We plan to study this particular subject in a follow-up project.

Connection to theories on the buckling transition to switch between faceted and bent structures.

We thank the reviewer for the constructive suggestion and add the following discussion to the supplementary information as extending discussion.

It is pleasing to see the present self-assembly of hollow rhombic dodecahedra as a manifestation of the quasi-equivalence principle—the association of identical building units into highly symmetric structures due to strong intermolecular H-bonding with high efficiency and no templates—although the current polyhedron size and free energy associated to pore formation might exceed the valid range of the principle.

Following Lidmar et al.'s arguments, the FoppeI–von Karman number is defined as $\gamma = E^{2D} R^2 / \kappa$, where E^{2D} is the 2D Young's modulus, R the facet size, and κ the bending rigidity. Larger γ favours buckled facets over spherical geometry. In our case, the 3D Young's modulus is estimated to be on the order of 1 GPa, so $E^{2D} \approx 4 \text{ Pa}\cdot\text{m}$ and $\kappa \approx 6\text{E-}18 \text{ J}$. Given $R \approx 1\text{E-}6 \text{ m}$, the γ is on the order of $1\text{E}6$. Such a large value indicates that the quasi-equivalence principle is not relevant to the formation of these shapes.

It is also surprising that, unlike the icosahedral geometry favoured by catanionic surfactants and many capsid proteins, the rhombic dodecahedral geometry is dominant possibly as a consequence of the in-plane rhombic lattice.

REVIEWERS' COMMENTS:

Reviewer #1 (Remarks to the Author):

The authors have addressed my concerns. I recommend publication of the revised manuscript.

Reviewer #2 (Remarks to the Author):

All my comments and concerns have been clearly addressed; The phase diagram makes this beautiful paper much easier to understand and I guess that this beautiful work- once published in the present form as I think it deserves, will have a large audience and trigger follow-up experimental investigation.

Response to Reviewer # 1 & 2

We thank the reviewers, who were both satisfied with the revised manuscript and recommended publication in its present form.